# On gauge freedom, conservativity and intrinsic dimensionality estimation in diffusion models

**Christian Horvat**
Department of Physiology
University of Bern
`christian.horvat@unibe.ch`

**Jean-Pascal Pfister**
Department of Physiology
Bern, Switzerland
`jeanpascal.pfister@unibe.ch`

## Abstract

Diffusion models are generative models that have recently demonstrated impressive performances in terms of sampling quality and density estimation in high dimensions. They rely on a forward continuous diffusion process and a backward continuous denoising process, which can be described by a time-dependent vector field and is used as a generative model. In the original formulation of the diffusion model, this vector field is assumed to be the score function (i.e. it is the gradient of the log-probability at a given time in the diffusion process). Curiously, on the practical side, most studies on diffusion models implement this vector field as a neural network function and do not constrain it be the gradient of some energy function (that is, most studies do not constrain the vector field to be conservative). Even though some studies investigated empirically whether such a constraint will lead to a performance gain, they lead to contradicting results and failed to provide analytical results. Here, we provide three analytical results regarding the extent of the modeling freedom of this vector field. Firstly, we propose a novel decomposition of vector fields into a conservative component and an orthogonal component which satisfies a given (gauge) freedom. Secondly, from this orthogonal decomposition, we show that exact density estimation and exact sampling is achieved when the conservative component is exactly equals to the true score and therefore conservativity is neither necessary nor sufficient to obtain exact density estimation and exact sampling. Finally, we show that when it comes to inferring local information of the data manifold, constraining the vector field to be conservative is desirable.

## 1 Introduction

Generative models generate data from noise. To do so, most generative models learn a mapping from the noisy latent space to the structured data space. Different mappings and different learning procedures lead to different models. For instance, for Normalizing Flows (Kobyzev et al., 2021), this mapping is bijective and trained on maximum likelihood. In contrast, for Generative Adversarial Networks (GANs) (Goodfellow et al., 2020) this mapping is not bijective, and the training objective is the Jensen-Shanon divergence between the model and data distribution. Recently, a new class of generative models - *diffusion models* - have shown tremendous success in various domains, even outperforming GANs regarding the visual fidelity of high-resolution images (Yang et al., 2022). Unlike a classical generative model that learns a single mapping from latent to data space, diffusion models *incrementally* add structure by an infinite series of *denoising* steps.

Mathematically, this can be described using stochastic differential equations such that starting from structured data at time $t = 0$, the data is unstructured at time $t = 1$. Crucially, this process can be reversed using the gradient of the true score function of the underlying stochastic process, that is, by using $s(\mathbf{x}, t) := \nabla \log p(\mathbf{x}, t)$ for $t \in [0, 1]$ and $\mathbf{x} \in \mathbb{R}^D$ where $p(\mathbf{x}, t)$ describes the density at time $t$. Here, we call *diffusion model* a neural network $s_\theta$ with parameters $\theta$ which approximates $s(\mathbf{x}, t)$. Even though $s(\mathbf{x}, t)$ is the gradient of the scalar function $\log p(\mathbf{x}, t)$, diffusion models are typically unrestricted in the sense that there is no guarantee that $s_\theta$ is the gradient of some scalar

function, that is there is no guarantee that $s_\theta$ is a *conservative vector field*. Several authors thus discussed the question whether $s_\theta$ should be conservative or not by construction (Salimans & Ho, 2021; Chao et al., 2023; Lai et al., 2023; Wenliang & Moran, 2022; Du et al., 2023; Cui et al., 2022). However, we argue that this is the wrong question to ask for the density estimation- and sampling ability of diffusion models. The right question to ask is if there is a functional freedom in diffusion models such that instead of learning the true score $s(\mathbf{x}, t)$, it is sufficient to learn a broader class of vector fields $s(\mathbf{x}, t) + r(\mathbf{x}, t)$. To do so, we will derive in section D non-trivial necessary and sufficient conditions for $r(\mathbf{x}, t)$ such that the corresponding generative model generates exact samples and learns the density exactly as well. We call this functional freedom *gauge freedom in diffusion models* [1].

As a direct consequence of this gauge freedom, we show that conservativity is neither necessary nor sufficient for exact density estimation and sampling. To the best of our knowledge, this is the first theoretical answer to the question of whether a diffusion model should be conservative by construction or not. Indeed, all previous work exclusively argued based on empirical evidence with contradicting results, as we will discuss shortly. Surprisingly, we also show that conservativity is sufficient when investigating local features of the data-manifold, such as local variability. In section 5, we present a method for estimating the intrinsic dimensionality of the data manifold by analyzing the local variability when approaching the data manifold. Overall, our findings can be summarized in terms of two takeaway messages to practitioners using diffusion models:

1. For density estimation or sampling, there is no need to constrain the diffusion model to be conservative. However, for exactness, the gauge freedom condition needs to be fulfilled.

2. For analyzing local features of the data manifold, such as the intrinsic dimensionality, a conservative diffusion is guaranteed to make the right conclusions.

Different authors studied the question of whether a diffusion model should be conservative or not. Salimans & Ho (2021) observed that, in terms of image generation, constraining $s_\theta$ to be conservative does lead to a similar performance as having no constraints on $s_\theta$. To ensure that $s_\theta$ is conservative, Salimans & Ho (2021) proposed to calculate the gradient of a scalar function, which requires an additional backward pass and is thus computationally more demanding than directly modeling a vector-valued $s_\theta$. As a result of this observation, it is widely accepted that $s_\theta$ can be unconstrained without losing much of generality (Song et al., 2021; Salimans & Ho, 2021; Yang et al., 2022; Liu et al., 2022; Zeng, 2023; Wenliang & Moran, 2022). However, in some application domains, using a consistent score function (Arts et al., 2023; Neklyudov et al., 2023) may be more suited.

Despite the findings of Salimans & Ho (2021), it is mathematically unsatisfactory to construct $s_\theta$, knowing that it is generally not consistent. This mathematical ghost is haunting the diffusion community, which is reflected in a surge of recent papers addressing this conflict (Chao et al., 2023; Lai et al., 2023; Wenliang & Moran, 2022; Du et al., 2023; Cui et al., 2022) . The corresponding conclusions are contradicting. For instance, while Wenliang & Moran (2022) report that a non-conservative $s_\theta$ learns a vector field that constrains the samples to be within the data-manifold and thus only little sampling improvements can be expected by enforcing $s_\theta$ to be conservative, . Thus, only little sampling improvements can be expected by enforcing $s_\theta$ to be conservative, Chao et al. (2023) observe that a non-conservative $s_\theta$ may lead to a degraded sampling performance. However, Chao et al. (2023) also observed that an unconstrained $s_\theta$ may enhance the density estimation ability. Therefore, Chao et al. (2023) and Cui et al. (2022) suggest implicitly enforcing conservativity by adding a penalty term to the usual objective function instead of explicitly modeling $s_\theta$ as a gradient of a scalar. This penalty term is $||\nabla s_\theta - \nabla s_\theta^T||_F$ where $\nabla s_\theta$ is the Jacobian of $s_\theta$ and $|| \cdot ||_F$ is the Frobenius norm of a matrix. As a vector field parametrized by a neural network is conservative if and only if the Jacobian is symmetric under some mild conditions (Im et al., 2016), this penalty indeed offers an incentive for $s_\theta$ to be conservative without losing the architectural freedom of unconstrained $s_\theta$.

On the practical side, Du et al. (2023) and Saremi (2019) give additional reasons in favor of conservative vector fields. Du et al. (2023) compose several likelihood models into a new one through multiplication, division, or summation. The latter refers to a mixture of distributions. However,

---

[1] In electromagnetics, the electric scalar potential, and the magnetic vector potential are not uniquely defined but enjoy some freedom, called gauge freedom, see chapter 10.1 in Griffiths (2005) or Abedi & Surace (2019) for a study on the gauge freedom in the context of non-linear filtering.

as discussed in Du et al. (2023), one can not use mixture composition without explicit likelihood functions, that is, without conservative vector fields in the case of diffusion models. In addition, conservative vector fields enable the use of more accurate numerical samplers such as Hamiltonian Monte Carlo (Duane et al., 1987; Neal, 2011). Surprisingly, Saremi (2019) showed that for an unconstrained $s_\theta$ to be conservative (thus being able to learn $s$ exactly), the weights of the first hidden layer must be parallel to the weights of the output layer. The author concludes that "the neural network is required to represent only one feature in its first hidden layer," providing a strong argument for explicitly constraining $s_\theta$ to be conservative.

## 2   NOTATIONS AND BACKGROUND

The high-level principle of diffusion models is to remove structure by adding noise incrementally in a way that can be reversed (Sohl-Dickstein et al., 2015; Song & Ermon, 2020). Recently, Song et al. (2021) unified different mathematical formulations of such models under the umbrella of stochastic differential equations (SDEs). For a comprehensive overview of the young history of diffusion models and alternative formulations, we refer to Yang et al. (2022). For our purposes, we adapt the notations and concepts of Song et al. (2021), which we will repeat in the following for convenience.

Let $\mathbf{x}_0 \in \mathbf{R}^D$ be random sample from the data distribution $p_0(\mathbf{x})$. Let us further assume that this random sample serves as an initialization for the following stochastic differential equation:

$$d\mathbf{x}_t = f(\mathbf{x}_t, t)dt + g(t)d\mathbf{w}_t \tag{1}$$

where $f : \mathbb{R}^D \times [0, 1] \to \mathbb{R}^D$ is a vector field, also known as *drift*, and $g : [0, 1] \to \mathbb{R}$ is the *diffusion coefficient* determining the magnitude of noise added at time $t$ as $\mathbf{w}_t$ is a $D-$dimensional *Brownian motion*. The stochastic process $\{\mathbf{x}_t\}_{t \in [0,1]}$ is referred to as *forward process*.

In principle, the drift $f$ and diffusion coefficient $g$ can be chosen almost arbitrarily. However, as one wants to ultimately reverse the process and generate new data, $f$ and $g$ need to be chosen such that the limiting distribution $p_1$ is known and can be easily sampled from.

**Sampling:** Surprisingly, Song et al. (2021) showed, based on results from Anderson (1982), that one can write down the reverse process $\{\mathbf{x}_t\}_{t \in [0,1]}$ with starting distribution $p_1$ and limiting distribution $p_0$ explicitly as a backward ODE using the gradient of $\log p$, $s(\mathbf{x}, t) := \nabla \log p(\mathbf{x}, t)$,

$$d\mathbf{x}_t = \tilde{f}(\mathbf{x}_t, t)dt \quad \text{with } \mathbf{x}_1 \sim p(\mathbf{x}, 1), \tag{2}$$

where $\tilde{f}(\mathbf{x}_t, t) := f(\mathbf{x}_t, t) - \frac{1}{2}g^2(t)s(\mathbf{x}_t, t)$. This allows one to sample new data from $p(\cdot, 0)$ by sampling from $p(\cdot, 1)$ and solving the ODE (2) backwards (that is from $t = 1$ to $t = 0$).

**Density estimation:** Moreover, equation (2) allows for estimating the density exactly by making use of the instantaneous change of variables formula (Chen et al., 2018),

$$\log p(\mathbf{x}_0, 0) = \log p(\mathbf{x}_1, 1) + \int_0^1 \nabla \cdot \tilde{f}(\mathbf{x}_t, t)dt \tag{3}$$

where $\nabla \cdot$ denotes the divergence operator. The latter applied on $\tilde{f}(\mathbf{x}_t, t)$ is the trace of the Jacobian of $\tilde{f}(\mathbf{x}_t, t)$ which can be efficiently estimated through automatic differentiation (Paszke et al., 2017) using the Skilling-Hutchinson trace estimator (Skilling, 1989; Hutchinson, 1990) ,

$$\nabla \cdot \tilde{f}(\mathbf{x}_t, t) = \text{Tr}\left(\nabla \tilde{f}(\mathbf{x}_t, t)\right) = \mathbb{E}_{\varepsilon \sim p(\varepsilon)}\left[\varepsilon^T \nabla \tilde{f}(\mathbf{x}_t, t)\varepsilon\right] \tag{4}$$

where, typically, $p(\varepsilon) = \mathcal{N}(\mathbf{0}, I)$, and $\nabla \tilde{f}(\mathbf{x}_t, t)$ is the Jacobian of $\tilde{f}(\mathbf{x}_t, t)$. Note that equation (3) depends on the whole trajectory $\{\mathbf{x}_t\}_{t \in [0,1]}$ drawn from (2).

Therefore, once we can learn $s(\mathbf{x}, t) = \nabla \log p(\mathbf{x}, t)$, we can sample new data through equation (2), and calculate the density explicitly through equation (3) (efficiently even in high-dimension thanks to equation (4), see Han et al. (2015)). Unexpectedly, to estimate $s(\mathbf{x}, t)$ using a neural network with parameters $\theta$, $s_\theta$, it is sufficient to estimate the conditional score $\log p_{0t}(\mathbf{x}_t | \mathbf{x}_0)$ - a procedure known as score matching (Hyvärinen & Dayan, 2005; Song & Ermon, 2019). With sufficient data and model flexibility, we have that $s_{\theta^*}(\mathbf{x}, t) = \nabla \log p(\mathbf{x}, t)$ for almost all $\mathbf{x}$ and $t$ where

$$\theta^* = \underset{\theta}{\text{argmin}} \mathbb{E}_{t \sim \mathcal{U}(0,1)}\left\{\lambda(t)\mathbb{E}_{\mathbf{x}_0}\mathbb{E}_{\mathbf{x}_t | \mathbf{x}_0}\left[||s_\theta(\mathbf{x}_t, t) - \nabla \log p_{0t}(\mathbf{x}_t | \mathbf{x}_0)||_2^2\right]\right\}. \tag{5}$$

The time $t$ is uniformly distributed on $[0, 1]$, $t \sim \mathcal{U}(0, 1)$, and $\lambda(t) : [0, 1] \to \mathbb{R}_{>0}$ is a positive weighting function. The expectation $\mathbb{E}_{\mathbf{x}_t|\mathbf{x}_0}$ is the expectation over $p_{0t}(\mathbf{x}_t|\mathbf{x}_0)$ such that given the drift and diffusion coefficient $f$ and $g$, respectively, we can sample from this conditional distribution efficiently at the one hand, and calculate $\nabla \log p_{0t}(\mathbf{x}_t|\mathbf{x}_0)$ explicitly on the other hand. However, data is typically limited, and $s_\theta$ is not arbitrarily flexible such that $s_\theta$ will not match the true score $s(\mathbf{x}, t)$ after training. Hence, constraining $s_\theta$ to be conservative or not can impact performance.

## 3 GAUGE FREEDOM FOR EXACT SAMPLING AND DENSITY ESTIMATION

To sample a data point with a diffusion model $s_\theta$, the initial value problem (IVP) (2) needs to be solved. Let $s_\theta$ be a vector field of the form

$$s_\theta(\mathbf{x}, t) = \nabla \log p(\mathbf{x}, t) + r_\theta(\mathbf{x}, t) \tag{6}$$

where $r_\theta : \mathbb{R}^D \times \mathbb{R} \to \mathbb{R}^D$ is summarizing the discrepancy between the learned vector field $s_\theta(\mathbf{x}, t)$ and the true score $s(\mathbf{x}, t) = \nabla \log p(\mathbf{x}, t)$. What is the gauge freedom of $r_\theta$ such that sampling with $s_\theta(\mathbf{x}, t)$ is equivalent to sampling with the true score $s(\mathbf{x}, t)$?

An equivalent description of the underlying ODE in (2) in terms of the corresponding density $p(\mathbf{x}, t)$ can be derived using the *Fokker-Planck equation* or also known as *Kolmogorov forward equation*. This equation, without diffusion term, is given by

$$\frac{\partial p(\mathbf{x}, t)}{\partial t} = -\nabla \cdot \left( \tilde{f}(\mathbf{x}, t) p(\mathbf{x}, t) \right), \tag{7}$$

see appendix D.1 in Song et al. (2021). This equation holds for every given point $\mathbf{x}$. The change in density along a path $\{\mathbf{x}_t\}_{t \in [0,1]}$ is given by the instantaneous change of variables formula from equation 3, see appendix A.2 in Chen et al. (2018) for details on how to derive the instantaneous change of variables formula from the Fokker-Planck equation. We will discuss the difference between equation (7) and equation (3) in more detail in section D .

Let the vector field corresponding to IVP (2) when using $s_\theta$ instead of $s$ be denoted by

$$\tilde{f}_\theta(\mathbf{x}, t) := f(\mathbf{x}, t) - \frac{1}{2} g^2(t) s_\theta(\mathbf{x}, t). \tag{8}$$

From that perspective, the above question can be reformulated as follows: What is the gauge freedom of $r_\theta$ such that the evolution of $p(\mathbf{x}, t)$ does not change when replacing $\tilde{f}$ with $\tilde{f}_\theta$? Suppressing the arguments to avoid clutter, standard calculus yields

$$\frac{\partial p}{\partial t} = -\nabla \cdot \left( \tilde{f} p \right) = -p \nabla \cdot \tilde{f} - \tilde{f}^T \nabla p = -p \left( \nabla \cdot \tilde{f} + \tilde{f}^T \nabla \log p \right) \tag{9}$$

where we have used $\nabla p = p \nabla \log p$ in the last step (we hence assume that $p \neq 0$). Now, replacing $\tilde{f}$ by $\tilde{f}_\theta$ in equation (9), we have that the density will not change whenever

$$\nabla \cdot \left( \frac{1}{2} g^2(t) r_\theta(\mathbf{x}, t) \right) + \left( \frac{1}{2} g^2(t) r_\theta(\mathbf{x}, t) \right)^T \nabla \log p(\mathbf{x}, t) = 0 \tag{10}$$

which is equivalent to

$$\nabla \cdot r_\theta(\mathbf{x}, t) + r_\theta(\mathbf{x}, t)^T \nabla \log p(\mathbf{x}, t) = 0 \tag{11}$$

as $g^2$ is typically independent of $\mathbf{x}$ and strictly positive. Therefore, whenever $r_\theta$ fulfills equation (11) for all $\mathbf{x} \in \mathbb{R}^D$ and $t \in [0, 1]$, we have that $s_\theta$ and $s$ will lead to the same samples and densities since the evolutions of the corresponding marginal probability distributions are the same.

The gauge freedom condition (11) yields a unique decomposition of any square-integrable (with respect to the measure induced by $p(\cdot, t)$) diffusion model $s_\theta$ into a sum of a conservative vector field and a remainder term satisfying equation (11), see theorem 1. This unique decomposition has some direct consequences, which we summarize in corollary 1. First, it shows that whenever the diffusion model $s_\theta$ is conservative, the remainder must vanish, $r_\theta(\mathbf{x}, t) = \mathbf{0}$. Second, it follows that exact sampling and density estimation is provided if and only if the conservative part is the true score, that is $s_\theta$ is given by equation (6)

**Theorem 1 (Orthogonal decomposition)** *Let $t \in [0,1]$. For any vector field $v \in L^2(p)$, there exists a unique conservative vector field $\nabla\phi \in L^2(p)$, and a unique vector field $r \in L^2(p)$ fulfilling the gauge freedom condition (11) such that*

$$v(\mathbf{x}, t) = \nabla\phi(\mathbf{x}, t) + r(\mathbf{x}, t). \tag{12}$$

**Corollary 1** *Let $v \in L^2(p)$ with unique decompositions $\nabla\phi$ and $r$ such that $v(\mathbf{x}, t) = \nabla\phi(\mathbf{x}, t) + r(\mathbf{x}, t)$, with $r$ satisfying the gaufe freedom condition (11).*

*(a) If $v(\mathbf{x}, t)$ is conservative, then it must hold that $r(\mathbf{x}, t) = \mathbf{0}$.*

*(b) $v(\mathbf{x}, t)$ provides exact density estimation and samples for the IVP 2 (replacing $s$ by $v$) if and only if $\nabla\phi(\mathbf{x}, t) = \nabla\log p(\mathbf{x}, t)$.*

Theorem 1 shows that the space of conservative vector fields in $L^2(p)$ is orthogonal to the space of vector fields fulfilling the gauge freedom condition in $L^2(p)$, see figute 1. This orthogonality provides a new intuition on the score matching loss in diffusion models. Let $s_\theta(\mathbf{x}, t) = \nabla\phi_\theta(\mathbf{x}, t) + r_\theta(\mathbf{x}, t) \in L^2(p)$, where $r_\theta$ satisfies the gauge freedom condition (11), then

$$\mathbb{E}\left[||s(\mathbf{x}, t) - s_\theta(\mathbf{x}, t)||_2^2\right] = \mathbb{E}\left[||s(\mathbf{x}, t) - \phi_\theta(\mathbf{x}, t)||_2^2\right] + \mathbb{E}\left[||r_\theta(\mathbf{x}, t)||_2^2\right]. \tag{13}$$

where the expectation is over $\mathbf{x} \sim p(\mathbf{x}, t)$. Thus score mathcing minimizes two terms. The first one on the right hand side of equation (13) is relevant (since when it is zero, correct sampling and density estimation can be obtained). The second term is irrelevant since it does not affect sampling and density estimation. However, for unconstrained $s_\theta$ this second term will be generally different from 0, see Saremi (2019), showing that unconstrained $s_\theta$ cannot match $s$ exactly.

**Remark 1** *A divergence-free remainder term is gauge freedom for the instantaneous change of variable formula derived in Chen et al. (2018), see section D in the appendix. Note, however, that this is not sufficient for exact sampling and density estimation as opposed to equation (11). The instantaneous change of variable formula derived in Chen et al. (2018) describes an ordinary differential equation for $p(\mathbf{x}_t, t)$, that is, how $p(\mathbf{x}_t, t)$ changes totally as a function of time. The Fokker-Planck equation (9), on the other hand, describes how $p(\mathbf{x}_t, t)$ changes partially as a function of time treating $\mathbf{x}_t$ as constant. The latter is a much stronger requirement ensuring that all vector fields with remainder term satisfying condition (11) correspond to the same marginals and thus stochastically equivalent sample paths.*

## 4 CONSERVATIVITY IS NEITHER NECESSARY NOR SUFFICIENT FOR EXACT DATA GENERATION AND LIKELIHOOD ESTIMATION

A direct consequence of the gauge freedom for diffusion models derived in section 3 is that conservativity is neither necessary nor sufficient for exact likelihood estimation or generating samples from the true data distribution, see figure 1. To substantiate the theoretical framework with an empirical illustration, in this section we construct a simple counter-example of a vector field $s_\theta$ which is not conservative but still satisfies the sufficient condition for exact density estimation and sampling, equation (11).

Let the target distribution be Gaussian with a diagonal covariance matrix. Then, by the additive closure of Gaussian distributions, the true score $s$ transforming a standard Gaussian to the target Gaussian must take the following form,

$$s(\mathbf{x}_t, t) = \nabla\log p(\mathbf{x}_t, t) = -\Sigma_t^{-1}\mathbf{x}_t, \text{ with } \Sigma_t^{-1} = \begin{pmatrix} \sigma_1^{-2}(t) & 0 \\ 0 & \sigma_2^{-2}(t) \end{pmatrix} \tag{14}$$

where $\sigma_1^2(t), \sigma_2^2(t) > 0$. Defining the remainder term as

$$r_\theta(\mathbf{x}_t, t) = R_t\mathbf{x}_t, \text{ with } R_t = \begin{pmatrix} 0 & -\sigma_1^2(t) \\ \sigma_2^2(t) & 0 \end{pmatrix}, \tag{15}$$

it is easy to verify the gauge freedom condition from equation (11): on the one hand, $r_\theta$ is divergence-free as the trace of the Jacobian $R_t$ is 0. On the other hand, we have that

$$r_\theta(\mathbf{x}_t, t)^T \cdot \nabla\log p(\mathbf{x}_t, t) = -\mathbf{x}_t^T R_t^T \Sigma_t^{-1}\mathbf{x}_t = -\mathbf{x}_t^T \begin{pmatrix} 0 & 1 \\ -1 & 0 \end{pmatrix} \mathbf{x}_t = 0. \tag{16}$$

Finally, note that $r_\theta$ cannot be conservative since the Jacobian is not symmetric (Schwarz theorem). Therefore, we have constructed a simple counter-example proving that the gauge freedom condition, equation (11), can be satisfied without the necessity of $s_\theta$ to be conservative. This example can be straightforwardly generalized for higher dimensions.

Conservativity is also not sufficient as, for example, $r_\theta(\mathbf{x}_t, t) = -s(\mathbf{x}_t, t)$ would reduce the backward ODE from equation (2) to be defined solely by the drift term $f$. For $f = 0$, such an ODE would only generate samples from the limiting distribution $p_1(\mathbf{x})$ as no dynamics are involved.

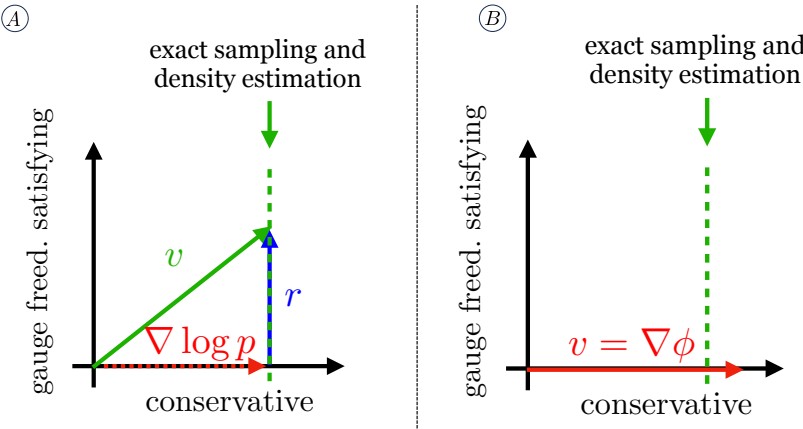

Figure 1: Every vector field $v \in L^2(p)$ can be orthogonally decomposed into a conservative vector field $\nabla\phi$ and a remainder term $r$ that satisfies the gauge freedom condition given by equation (11). (A) Exact sampling and density estimation is obtained when the conservative component $\nabla\phi$ of the vector field $v$ is equal to the true score (i.e. $\nabla\phi = \nabla\log p$) - which is the case for all the points on the green dashed line. So $v$ does not need to be conservative. (B) Even if $v$ is conservative, it is not sufficient to guarantee exact sampling and density estimation since it may be different than the true score.

## 5 A CONSERVATIVE VECTOR FIELD IS DESIRED FOR EXACT LOCAL INFORMATION

In this section, we show how to estimate the intrinsic dimensionality of the data manifold whenever $s_\theta$ matches the true score $s$. Additionally, we provide empirical evidence that with a non-conservative vector field, the ID is not estimated correctly while using the derived method we can estimate the ID correctly if $s_\theta$ is constrained to be conservative (not necessarily matching $s$ exactly). This suggests that constraining the diffusion model to be conservative should be preferred for inferring local information.

A sample from a diffusion model is the solution to an initial value problem, see equation (2). We denote the solution of this IVP as $\phi_t(\mathbf{x}_1)$ where $\mathbf{x}_1 \sim p(\mathbf{x}, 1)$. Note that this solution is unique whenever $f$ and $g$ are globally Lipschitz in both state and time Øksendal (2003). How does this solution depend on the initial value $\mathbf{x}_1$? Let $s > 0$ and $\varepsilon \sim \mathcal{N}(0, I)$, then we have that

$$\phi_t(\mathbf{x}_1 + s\varepsilon) = \phi_t(\mathbf{x}_1) + s\frac{\partial\phi_t(\mathbf{x}_1)}{\partial\mathbf{x}_1}\varepsilon + \mathcal{O}(s^2)$$

and hence for $s \to 0$,

$$\frac{\phi_t(\mathbf{x}_1 + s\varepsilon) - \phi_t(\mathbf{x}_1)}{s} \xrightarrow{d} \mathcal{N}(0, Y(\mathbf{x}_1, t)Y(\mathbf{x}_1, t)^T) \tag{17}$$

where we have defined $Y(\mathbf{x}_1, t) = \frac{\partial\phi_t(\mathbf{x}_1)}{\partial\mathbf{x}_1}$, and $\xrightarrow{d}$ denotes convergence in distribution. Equation (17) means that a diffusion model maps locally a Gaussian distribution into a Gaussian distribution. Therefore, we can relate local information on the manifold, such as directions and strengths of

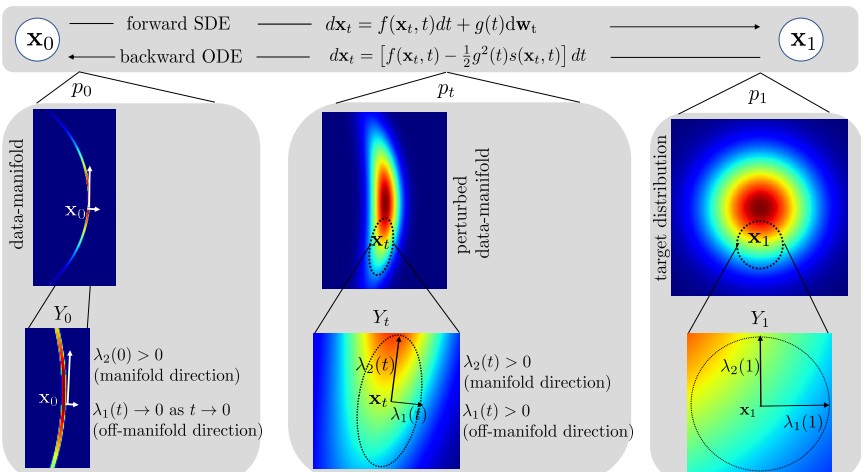

Figure 2: Intuition of how the singular values of $Y(\mathbf{x}_1, t) = \frac{\partial \phi_t(\mathbf{x}_1)}{\partial \mathbf{x}_1}$ evolve over time for a low-dimensional data-manifold. The singular value in the manifold direction will saturate, while the singular values in the off-manifold direction will tend to 0 (bottom left).

variability, to the singular vectors and singular values of $Y(\mathbf{x}_1, 0)$, respectively, see figure 2 . A similar result to equation (17) was also observed for normalizing flows (Cunningham et al., 2022; Horvat & Pfister, 2022), and was exploited by Horvat & Pfister (2022) to use normalizing flows for estimating the intrinsic dimensionality of low-dimensional manifolds. In the following, we want to estimate the ID similarly using diffusion models. For the sake of brevity, from now on, we drop the dependence of $Y(\mathbf{x}_1, t)$ on $\mathbf{x}_1$ and set $Y_t := Y(\mathbf{x}_1, t)$.

Figure 2 serves as an illustration of the main idea. Starting from a low-dimensional manifold, an arc embedded in $\mathbb{R}^2$, with a density $p_0$, we gradually transform the data-density to a standard Gaussian $p_1$. To sample a new data point, we first sample $\mathbf{x}_1$, and to analyze how a vicinity of $\mathbf{x}_1$ evolves through the backward diffusion, we can study how the singular values of $Y_t$ change as a function of time (bottom row). Crucially, since the data lives on a low-dimensional manifold, the singular value of $Y_t$ associated with the off-manifold directions must approach zero when $t \to 0$. At the same time, the other singular value will converge to some fixed value. Indeed, the sensitivity to the initial condition of the backward denoising process is much larger along the "on-manifold" direction than on the "off-manifold" direction. Therefore, if we can study how the singular values of $Y_t$ evolve as a function of time, the number of saturating singular values will correspond to the true intrinsic dimensionality.

Unfortunately, we cannot access $Y_t$. However, a standard result from the study of ODE is that $Y_t$ is the solution of

$$dY_t = \nabla \tilde{f}(\phi_t(\mathbf{x}_1), t) Y_t dt, \quad Y_0 = I \tag{18}$$

where $\nabla \tilde{f}(\phi_t(\mathbf{x}_1), t)$ is the Jacobian of $\tilde{f}(\mathbf{x}, t)$ evaluated at $(\phi_t(\mathbf{x}_1), t)$, see Teschl (2012). This description of $Y_t$ allows us to express the singular values of $Y_t$ in terms of the eigenvalues of $\nabla \tilde{f}(\phi_t(\mathbf{x}_1), t)$ which we can calculate in practice when we approximate the gradient of the true score $\nabla \log p(\mathbf{x}, t)$ using a diffusion model $s_\theta(\mathbf{x}, t)$. In the supplementary materials, we prove the following theorem:

**Theorem 2** *Let the data distribution $p(\cdot, 0)$ be supported on a low-dimensional manifold $\mathcal{M}$ of dimension $d$ embedded in $\mathbb{R}^D$. Let $s_\theta(\mathbf{x}, t) \in L^2(p)$ be a diffusion model trained on data from $p(\cdot, 0)$ providing exact samples and density estimation. Let $P_t(\mathbf{x}_1) := Y(\mathbf{x}_1, t) Y(\mathbf{x}_1, t)^T$ have smooth eigenvalues in $t$ for all $\mathbf{x}_1 \in \mathbb{R}^D$ where $Y(\mathbf{x}_1, t) = \frac{\partial \phi_t(\mathbf{x}_1)}{\partial \mathbf{x}_1}$. Suppose that there exists an $\varepsilon > 0$ such that $[P_t(\mathbf{x}_1), \nabla \tilde{f}_\theta(\phi_t(\mathbf{x}_1), t)] = 0$ for all $t \in [0, \varepsilon]$, that is, $P_t(\mathbf{x}_1)$ and $\nabla \tilde{f}_\theta(\phi_t(\mathbf{x}_1), t)$ commute for all $t \in [0, \varepsilon]$. Then, we have that*

$$s_\theta \text{ is conservative} \implies \text{rank} \left[ \exp \left( \nabla \tilde{f}(\mathbf{x}_0, 0) \right) \right] = \text{rank} \left[ \exp \left( \nabla \tilde{f}_\theta(\mathbf{x}_0, 0) \right) \right] = d, \tag{19}$$

*where* $\mathbf{x}_0 = \phi_0(\mathbf{x}_1)$.

Theorem (2) shows that the intrinsic dimensionality can be estimated using the rank of the Jacobian of $\tilde{f}_\theta$. In the following, we will empirically confirm this.

**Remark 2** *If both $P_t$ and $\nabla \tilde{f}_{\theta,t}$ are diagonalizable on the one hand, and $P_t$ and $\nabla \tilde{f}_{\theta,t}$ have the same eigenvectors on the other hand, we have that indeed $[P_t, \nabla \tilde{f}_{\theta,t}] = 0$. For sufficiently small $\varepsilon$, the eigenvectors of $\nabla \tilde{f}_{\theta,t}(\mathbf{x})$ will align with the normal and tangent space of the manifold, see discussion in Wenliang & Moran (2022). Also, (Permenter & Yuan, 2023) support this hypothesis as they show that denoising is approximately projecting close to the data manifold. As also the singular vectors of $P_t$ will align with the normal and tangent space, the assumption that $[P_t(\mathbf{x}_1), \nabla \tilde{f}_\theta(\phi_t(\mathbf{x}_1), t)] = 0, \forall t \in [0, \varepsilon]$, for a sufficiently small $\varepsilon$ is reasonable.*

**Intrinsic dimensionality estimation using diffusion models:** We consider a $2-$dimensional Gaussian embedded in $\mathbb{R}^5$ as a toy example and a proof of concept. Thus, the intrinsic dimensionality of the data-manifold is 2. We train a conservative and non-conservative diffusion model using $f(\mathbf{x}, t) = 0$ and $g(t) = 25^t$ as drift and diffusion coefficient, respectively. The non-conservative diffusion model is simply an unconstrained neural network $s_\theta(\mathbf{x}, t) = \psi_\theta$ where $\psi_\theta : \mathbb{R}^5 \times \mathbb{R}_{>0} \to \mathbb{R}^5$. The conservative version is $s_\theta(\mathbf{x}, t) = \nabla ||\psi_\theta(\mathbf{x}, t)||_2^2$ as suggested by Du et al. (2023). In figure 3 A, we see the evolution of the singular values as stated in theorem 2 as a function of time in log-log scale. Each color stands for 1 of a total of 5 singular values. If we use a conservative vector field (left plot), the singular values evolve as predicted; that is, 2 of them saturate, whereas the remaining 3 diverge. All 5 singular values saturate for the non-conservative vector field, and the intrinsic dimensionality cannot be estimated using the singular values of $Y_1$. Although we only show the trajectories for one representative sample, we observe the same behavior across different samples.[2]

In figure 3 B, we show that our method scales with increasing embedding and intrinsic dimension. Our method perfectly matches the true intrinsic dimension for a sphere with dimension $D/2 - 1$ embedded in $D$ for different values of $D$. We conduct more experiments in the supplementary on different manifolds (spheres, tori, swiss rolls) with different embedding dimensions and observe that the results do not change: a conservative $s_\theta$ can estimate the intrinsic dimension exactly whereas a non-conservative does not - even if we increase the number of parameters used for $s_\theta$ or add a penalty term enforcing conservativity by symmetrizing the Jacobian of $s_\theta$ as suggested by Chao et al. (2023) and Cui et al. (2022).

Note that also Wenliang & Moran (2022) and Batzolis et al. (2022) estimate the intrinsic dimension using diffusion models. However, Wenliang & Moran (2022) does not come with any theoretical guarantee for estimating $d$ correctly, and Batzolis et al. (2022) does not estimate the ID correctly for spheres. Besides, our main motivation is to discuss the gauge freedom and conservativity question and their importance for correctly inferring local information. We did not focus on developing a state-of-the-art ID estimator, which is why we leave a thorough comparison of recent ID estimators based on neural networks (Horvat & Pfister, 2022; Batzolis et al., 2022; Wenliang & Moran, 2022; Tempczyk et al., 2022; Mohan et al., 2019) for the future.

## 6   DISCUSSION

In this paper, we have argued that instead of asking whether a diffusion model should be a conservative vector field (as required by the original theory) or not (as usually done in practice), a better question to ask is if there exists a greater class of diffusion models without sacrificing exactness in both density estimation and sampling ability. Indeed, we have demonstrated theoretically that diffusion models enjoy a gauge freedom for data synthesis and density estimation. As a direct consequence of this gauge freedom, we have shown that conservativity is neither necessary nor sufficient for exact density estimation or perfect sampling. To the best of our knowledge, this is the first theoretical answer to the conservity question, which was previously only addressed empirically with contradicting and unsatisfying results. Our theory also provides new intuition on the score-matching objective and confirms previous results that an unconstrained diffusion model will likely not learn the true score exactly.

---

[2]We added some slack to the curves for better display as some overlap.

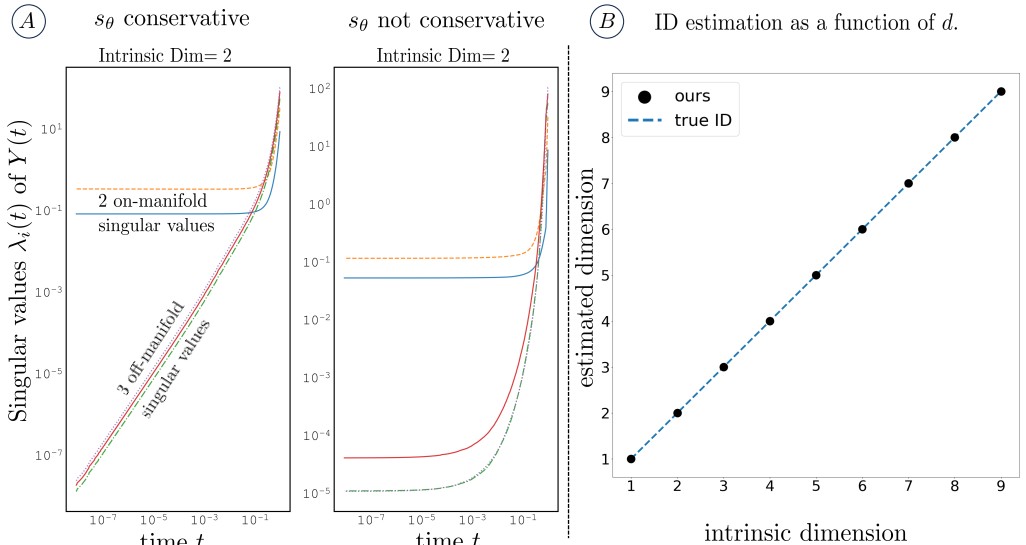

Figure 3: **A:** Singular values of $Y_t$ as predicted by lemma 1 in the appendix for $s_\theta$ conservative (left) and non-conservative (right). Each color represents one singular value (5 in total as the embedding dimension is 5). **B:** Intrinsic dimensionality estimation of sphere with dimension $d = D/2 - 1$ embedded in $D$ for different values of $d$.

In practice, enforcing the gauge freedom conditions may be challenging since, to do so, one would need access to the true score function. However, since time-continuous diffusion models are trained to learn the transitional probability function $p_{0t}(\mathbf{x}_t|\mathbf{x}_0)$ for all times $t$ and data points $\mathbf{x}_0$, see equation (5), one could add penalty terms to enforce the gauge freedom conditions accordingly. We leave this exploration for the future.

Finally, we derived in the appendix lemma 1, which relates the singular values of $Y(t, \mathbf{x}_1)$ (which are unknown) to the singular values of $\tilde{f}_\theta$ (which can be calculated), for conservative vector fields only. As the singular values of $Y_t$ describe how a small neighborhood of the initial value $\mathbf{x}_1$ evolves when applying the sample generating ODE, we have used this information to estimate the intrinsic dimensionality of the data-manifold. We have seen empirically that only if $s_\theta$ is indeed conservative, we obtain the right behavior as predicted by theorem 2 . Though one key assumption of theorem 2 is strong, namely that the commutator of $Y_t Y_t^T$ and $\nabla \tilde{f}_\theta$ vanishes sufficiently close to the manifold, we demonstrated on different manifolds that, nevertheless, the singular values behave as predicted, and the true ID can be estimated. Therefore, we hypothesize that, indeed, the eigenvectors of $\nabla \tilde{f}_\theta$ and $Y_t Y_t^T$ align close to the data manifold. Finally, the intrinsic dimensionality should be also estimated correctly if the remainder term $r_\theta$ of $s_\theta = s + r_\theta$ fulfills the gauge freedom condition. However, as discussed, this is difficult to ensure in practice. Relaxing the conditions of theorem 2 to accommodate for the general case is an interesting direction to pursue and might provide new insights on the gauge freedom condition.

As a takeaway message, when using diffusion models for data synthesis or density estimation, conservativity is neither necessary nor sufficient, but the gauge freedom condition from equation (11) is necessary for the remainder term $r_\theta$ when the diffusion model is expressed as $s_\theta = s + r_\theta$. However, when one is interested in inferring local information of the data-manifold using diffusion models, we recommend working with a conservative vector field such that the right conclusion can be made.

## ACKNOWLEDGMENTS

We express our sincere gratitude to the anonymous reviewers for their meticulous examination of our manuscript and their valuable recommendations. We would like to extend special acknowledgment to Reviewer h3ie, whose insightful observations regarding gauge freedom significantly contributed to the formulation of theorem 1 and corollary 1., see `https://openreview.net/forum?id=92KV9xAMhF&noteId=jwbeNMqn9r` for details.

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

## A  PROOF OF THEOREM 1

Let $v \in L^2(p)$, that is

$$||v||^2_{L^2(p)} := \mathbb{E}_{\mathbf{x} \sim p(\mathbf{x},t)} \left[ v^2(\mathbf{x}) \right] = \int_{\mathbb{R}^D} v^2(\mathbf{x}) p(\mathbf{x},t) d\mathbf{x} < \infty. \tag{20}$$

Note that $L^2(p)$ is a Banach space with a scalar product inducing above norm. This scalar product allows to define the orthogonal complement of the subspace of conservative vector fields in $L^2(p)$. As we will see below, this complement is exactly the space of vector fields fulfilling the gauge freedom condition. Finally, this complement is also a *closed* subset[3] and thus, Banachs projection theorem guarantees the desired unique decomposition, see theorem 3.6.6 Debnath & Mikusinski (2005).

What is left to show is the aforementioned orthogonality. Let $\nabla \phi \in L^2(p)$, and $r \in L^2(p)$ fulfilling the gauge freedom condition (11). We have that

$$\begin{aligned} \langle \nabla \phi | r \rangle_{L^2(p)} &= \mathbb{E}_{\mathbf{x} \sim p(\mathbf{x},t)} \left[ \nabla \phi(\mathbf{x},t) r(\mathbf{x},t) \right] \\ &= \int_{\mathbb{R}^D} \nabla \phi(\mathbf{x},t) r(\mathbf{x},t) p(\mathbf{x},t) d\mathbf{x} \\ &= - \int_{\mathbb{R}^D} \phi(\mathbf{x},t) (\nabla \cdot r(\mathbf{x},t) + r(\mathbf{x},t)^T \nabla \log p(\mathbf{x},t)) d\mathbf{x} \\ &\stackrel{(11)}{=} 0 \end{aligned} \tag{21}$$

where in the second to last equation we have use the integration by parts formula. Thus, $\nabla \phi$ is orthogonal to $r$ in the Hilbert space $L^2(p)$. □

---

[3]A proof of this standard result from the study of Hilbert spaces can be found in Debnath & Mikusinski (2005), theorem 3.6.2.

# B  PROOF OF COROLLARY 1

Corollary (a) is a direct consequnce of the uniqueness of the decomposition. One direction of corollary (b) is shown in the beginning of section (3). There, we have shown that if the conservative part of $v$ indeed matches the true score, then $v$ provides exact samples for the IVP 2.

Now, we assume that $v$ provides exact samples for the IVP 2. Thus, the difference between $v$ and the true score needs to fulfill the Gauge freedom condition (11), that is

$$\nabla\phi(\mathbf{x}, t) - \nabla\log p(\mathbf{x}, t) + r(\mathbf{x}, t) \tag{22}$$

fulfills equation (11). However, $r$ already fulfills equation (11), and conservative vector fields are orthogonal to vector fields fulfilling equation (11). Therefore, the conservative part $\nabla\phi(\mathbf{x}, t) - \nabla\log p(\mathbf{x}, t)$ needs to vanish. With other words, it must hold that $\nabla\phi(\mathbf{x}, t) = \nabla\log p(\mathbf{x}, t)$ which was left to show. $\square$

# C  PROOF OF THEOREM 2

As we assume that $s_\theta$ is conservative and yields exact sampling and density estimation, corollary 1 implies that $s_\theta = s$. Thus, we have that $\tilde{f}_\theta = \tilde{f}$. What is left to show is that the rank of the matrix exponential $\exp\left(\nabla\tilde{f}(\mathbf{x}, t)\right)$ converges to $d$. To do so, we will relate the singular values of $P_t$ with the eigenvalues of $\nabla\tilde{f}(\mathbf{x}, t)$ through lemma 1 . Note that the rank of $\lim_{t\to 0} Y_t Y_t^T$ where $Y_t(\mathbf{x}_1) = \frac{\partial\phi_t(\mathbf{x}_1)}{\partial\mathbf{x}_1}$ must be $d$ as $\phi_t(\mathbf{x}_1)$ is the solution to the IVP from equation (2), and $P_0 := Y_0 Y_0^T$ defines the local variability on the manifold (which is $d$-dimensional) see equation (17). As the rank of $P_0$ is given by the number of non-zero singular values of $Y_0$, we will see how the aformentioned relation allows us to estimate $d$ by the number of non-exploding eigenvalues of $\lim_{t\to 0}\nabla\tilde{f}(\mathbf{x}, t)$, or equivalently: the rank of $\exp\left(\nabla\tilde{f}(\mathbf{x}, 0)\right)$.

**Lemma 1** *With the same assumptions as in theorem 2, let $\nabla\tilde{f}(\phi_t(\mathbf{x}_1), t)$ have eigenvalues $\mu_1(t) \leq \cdots \leq \mu_D(t)$, then the eigenvalues $0 \leq \lambda_1(t) \leq \cdots \leq \lambda_D(t)$ of $P_t(\mathbf{x}_1)$ are given by*

$$\lambda_i(t) = \lambda_i(\varepsilon)\exp\left(2\int_t^\varepsilon \mu_i(s)\mathrm{ds}\right) \tag{23}$$

*for all $t \in [0, \varepsilon]$*

*Proof of lemma 1:*  The singular values of $Y_t(\mathbf{x}_1)$ are given by the eigenvalues of $Y_t^T(\mathbf{x}_1)Y_t(\mathbf{x}_1)$. We simply write $Y_t$ instead of $Y_t(\mathbf{x}_1)$ in the following. Note that $P_t = Y_t Y_t^T$ has the same eigenvalues as $Y_t^T Y_t$ (but not necessarily the same eigenvectors). Let $\mathbf{p}_i$ be an eigenvector of $P_t$ with eigenvalue $\lambda_i \neq 0$ (see lemma 2), that is $P_t\mathbf{p}_i = \lambda_i\mathbf{p}_i$. Then, taking the time derivative on both sides of the eigenvector equation, we get

$$\dot{P}_t\mathbf{p}_i + P_t\dot{\mathbf{p}}_i = \dot{\lambda}_i\mathbf{p}_i + \lambda_i\dot{\mathbf{p}}_i \tag{24}$$

Note that every symmetric matrix has an eigenvector decomposition consisting of orthonormal eigenvectors. In this context, $\dot{\mathbf{p}}_i$ is either orthogonal to $\mathbf{p}_i$ (hence an eigenvector of $P$) or $\dot{\mathbf{p}}_i = \mathbf{0}$. In both cases we have that $\langle\dot{\mathbf{p}}_i, \mathbf{p}_i\rangle = 0$. Therefore, if we multiply both sites of equation (24) with $\mathbf{p}_i^T$ from the left, we have that

$$\langle\mathbf{p}_i|\dot{P}\mathbf{p}_i\rangle = \dot{\lambda}_i. \tag{25}$$

Note that

$$\dot{P}_t = \dot{Y}_t Y_t^T + Y_t\dot{Y}_t^T = \nabla\tilde{f}P_t + P_t\nabla\tilde{f} \tag{26}$$

since $\dot{Y}_t = \nabla\tilde{f}_t Y_t$ where $\nabla\tilde{f}_t := \nabla\tilde{f}(\phi_t(\mathbf{x}_1), t)$, and thus for the transpose holds $\dot{Y}_t^T = Y_t^T\nabla\tilde{f}_t^T = Y_t^T\nabla\tilde{f}_t$ (note that $\nabla\tilde{f}_t$ is symmetric as $s_\theta$ is conservative by assumption). Introducing the commutator $[P_t, \nabla\tilde{f}_t] := \nabla\tilde{f}_t P_t - P_t\nabla\tilde{f}_t$, which is 0 for all $t \in [0, \varepsilon]$ by assumption, we can further simplify above equation for $t \in [0, \varepsilon]$

$$\dot{P}_t = [P, \nabla\tilde{f}_t] + 2P_t\nabla\tilde{f}_t = 2P_t\nabla\tilde{f}_t. \tag{27}$$

Note that $[P_t, \nabla \tilde{f}_t] = 0$ implies that $\nabla \tilde{f}_t p_i$ is an eigenvector of $P_t$ as $P_t \nabla \tilde{f}_t \mathbf{p}_i = \nabla \tilde{f}_t P_t \mathbf{p}_i = \lambda_i \nabla \tilde{f}_t \mathbf{p}_i$. If $\dot{\lambda}_i \neq 0$ in equation (25), then we have that $\nabla \tilde{f}_t \mathbf{p}_i = \mu_i \mathbf{p}_i$ for some $\mu_i \in \mathbb{R} \backslash \{0\}$ as otherwise $\nabla \tilde{f}_t \mathbf{p}_i = \mu_i \mathbf{p}_j$ for some $j \neq i$ and hence $\langle \mathbf{p}_i | \dot{P}_t \mathbf{p}_i \rangle = 2\langle \mathbf{p}_i | P_t \nabla \tilde{f}_t \mathbf{p}_i \rangle = 2\mu_i \lambda_j \langle \mathbf{p}_i | \mathbf{p}_j \rangle = 0$ which is a contradiction to $\dot{\lambda}_i \neq 0$.

If $\dot{\lambda}_i = 0$, however, then we must have for all $i$ that $\nabla \tilde{f}_t \mathbf{p}_i$ is an element in the space spanned by all eigenvectors except $\mathbf{p}_i$. In other words, $\nabla \tilde{f}_t$ is a change-of-basis with a permutation matrix as a change-of-basis matrix. However, such a transformation cannot be symmetric which we have assumed for $\nabla \tilde{f}_t$.

Therefore, we have that $\dot{\lambda}_i \neq 0$ and $\nabla \tilde{f}_t \mathbf{p}_i = \mu_i \mathbf{p}_i$.

Then,
$$\dot{P}_t \mathbf{p}_i = 2 P_t \nabla \tilde{f}_t \mathbf{p}_i = 2\lambda_i \mu_i \mathbf{p}_i. \tag{28}$$

Finally, inserting this into equation (25) we have that

$$\dot{\lambda}_i = \langle \mathbf{p}_i | \dot{P}_t \mathbf{p}_i \rangle$$
$$\Longleftrightarrow \quad \dot{\lambda}_i = \langle \mathbf{p}_i | 2\lambda_i \mu_i \mathbf{p}_i \rangle$$
$$\Longleftrightarrow \quad \dot{\lambda}_i = 2\mu_i \lambda_i$$
$$\Longleftrightarrow \quad \frac{\dot{\lambda}_i}{\lambda_i} = 2\mu_i$$
$$\Longleftrightarrow \quad \frac{\mathrm{d}}{\mathrm{dt}} \ln(\lambda_i) = 2\mu_i$$
$$\Longleftrightarrow \quad \lambda_i(t) = \lambda_i(\varepsilon) \exp\left( 2 \int_t^\varepsilon \mu_i(s) ds \right) \tag{29}$$

Note that we for the third step, we need that $\lambda_i \neq 0$ which we proof below in lemma 2. This ends the proof. $\square$

**Lemma 2** *The eigenvalues $\lambda_i(t)$ are non-zero for all $t \geq 0$.*

**Proof:** Liouvilles formula for the determinant of the matrix solution, see lemma 3.11 in Teschl (2012), to the ODE

$$dY_t = \nabla \tilde{f}(\phi_t(\mathbf{x}_0), t) Y_t dt$$
$$Y_0 = I \tag{30}$$

states that

$$\det Y_t = \det Y_0 \exp\left( \int_0^t \mathrm{Tr}\left( \nabla \tilde{f}(\phi_t(\mathbf{x}_0), t) \right) \mathrm{dt} \right). \tag{31}$$

The trace of $\nabla \tilde{f}(\phi_t(\mathbf{x}_0), t)$ is given by $\sum_i \mu_i(t)$, and the determinant of $Y_t$ by $\Pi_i \lambda_i(t)$. The right-hand side is always non-zero. Therefore, each factor on the left-hand side is non-zero. This is what we wanted to show. $\square$

Finally, we finish the proof of theorem 2. The rank of $P_t$ is $d$ by the characterisation of $P_t$ through equation 17. On the other hand, the rank is given by the number of non-zero eigenvalues of $P_t$. These eigenvalues can be calculated using the eigenvalues of $\nabla \tilde{f}_t$, see lemma 1 . Thus, $(D - d)$ eigenvalues of $\nabla \tilde{f}_t$ must converge to $-\infty$ for $t \to 0$ which corresponds to the rank of $\exp \nabla \tilde{f}_{\theta,0}$ which is what we wanted to show. $\square$.

## D    IF EXACT SAMPLING IS PROVIDED, HELMHOLTZ DECOMPOSABILITY IS SUFFICIENT FOR EXACT DENSITY ESTIMATION

In the previous section, we have derived a gauge freedom for diffusion models expressed in equation (10). By initially considering the ODE formulation of the sampling procedure, we have exploited the equivalent description of sample trajectories in terms of the underlying marginal probability

densities given by the Fokker-Planck equation. The close relation between sampling and density estimation is not surprising as evaluating the density, see equation (3), requires knowledge of the entire sample trajectory. In this section, we show that if the model generates exact samples, Helmholtz decomposibility is sufficient for exact density estimation.

Let $s_\theta$ be given by equation (6) with $r_\theta(\mathbf{x}, t)$ being a rotation field (that is a vector field with $\nabla \cdot r_\theta(\mathbf{x}, t) = 0$). Replacing the true score by $s_\theta(\mathbf{x}, t)$ for evaluating the model likelihood in equation (3), will lead to the same likelihood because

$$\text{Tr}\left(\nabla s_\theta(\mathbf{x}_t, t)\right) = \text{Tr}\left(\nabla^2 \log p(\mathbf{x}_t, t)\right) + \text{Tr}(\nabla r_\theta(\mathbf{x}_t, t)) = \text{Tr}\left(\nabla^2 \log p(\mathbf{x}_t, t)\right) + \nabla \cdot r_\theta(\mathbf{x}_t, t)$$

which results in $\text{Tr}\left(\nabla^2 \log p(\mathbf{x}_t, t)\right)$ as the trace of the Jacobian of $r_\theta$ is equal to the divergence of $r_\theta$ which is 0 for all rotation fields. Therefore, for a given path $\{\mathbf{x}_t\}_{t \in [0,1]}$, the diffusion model $s_\theta$ as defined in equation (6) and the true score $s$ yield the same density when using equation (3) to estimate $p_0(\mathbf{x}_0)$, no matter how close $s_\theta$ is to the true score.

# E  INTRINSIC DIMENSIONALITY ESTIMATION

As mentioned at the end of Section 5.1 of the main text, we perform more experiments for estimating the intrinsic dimensionality.

For the non-conservative diffusion model, we simply use a standard feed-forward neural network where we first embed the data into 100 dimensions and linearly transform it followed by a non-linearity (first step). Further, we embed the resulting features into 200 dimensions, again linearly transform it followed by a non-linearity, and finally project back into the data dimensions (second step). We embed the time into 100 dimensions using a Gaussian-Fourier projection and add these embeddings to the features after the first step. The conservative version additionally takes the gradient of the corresponding L2-norm with respect to the inputs.

In figure 5 and 4 we show the evolution of the singular values (in log-log scale) as a function of time for the Swiss Roll, Sphere, and Torus embedded in $D = 3$ on the left and for embedding dimension $D = 5$ on the right, respectively. On each side, we show the evolution for both a conservative and not conservative diffusion model $s_\theta$. The number of lines corresponds to the embedding dimensions $D$ as this is the number of singular values of $Y$. We can see that for $s_\theta$ conservative, always 2 of in total $D$ singular values saturate when approaching the manifold (that is when $t \to 0$). However, the remaining singular values do not saturate and tend to $-\infty$, that is the singular values tend to $0$ (confirming the intuition from the main text). For $s_\theta$ not conservative, however, all singular values saturate showing that $s_\theta$ does not behave as predicted close to the manifold. Even if we add a penalty term the Jacobian enforcing symmetry and thus conservativity, as suggested in Chao et al. (2023), we observe the same scaling behavior.

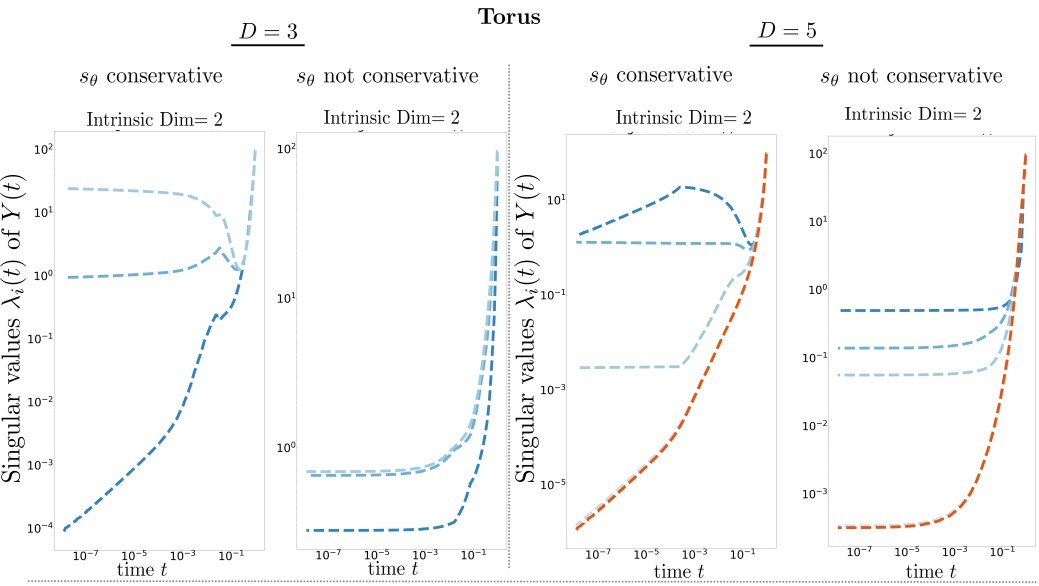

Figure 4: Singular values trajectories of as torus for different embedding dimensions ($D = 3$ and $D = 5$). We show the evolution of both a conservative and not conservative diffusion model.

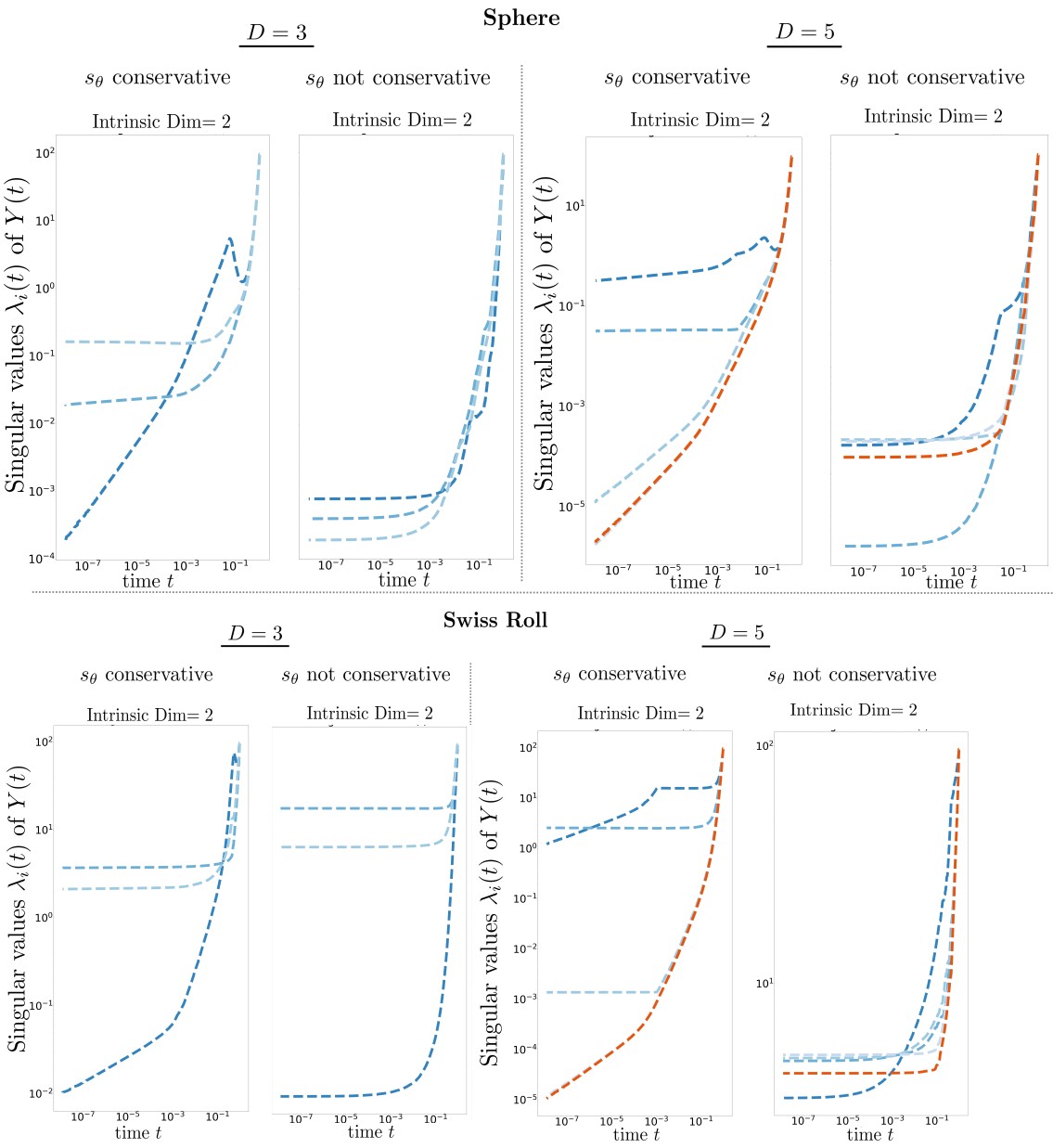

Figure 5: Singular values trajectories of the Swiss Roll and sphere for different embedding dimensions ($D = 3$ and $D = 5$). We show the evolution of both a conservative and not conservative diffusion model.

