# OpenReview forum: "On gauge freedom, conservativity and intrinsic dimensionality estimation in diffusion models"
_ICLR.cc/2024/Conference — ICLR 2024 poster_

### Official Review · Reviewer_pZCv · 2023-10-29

**Soundness:** 2 fair
**Presentation:** 2 fair
**Contribution:** 2 fair
**Rating:** 6
**Confidence:** 3

**Summary:**

Score-based generative models achieve state-of-the-art performance in image generation tasks, and are based on fitting a model to the gradient of the log likelihood of blurred data. As such, it would be natural to fit models which are themselves conservative (the gradient of a function), but in practice this constraint is not enforced at all, and instead general vector fields, parametrized by a deep model, are used. This discrepancy leads to a natural question that is addressed in this work - how does learning a possibly non-conservative vector field affect the performance of score-based generative models? The authors derive an equation that precisely addresses this question (they refer to this as a gauge invariance), and show that it can be obeyed by both conservative and non-conservative vector fields. They then consider the problem of intrinsic dimension estimation and show that a conservative vector field -- under some technical assumptions -- is sufficient to estimate the intrinsic dimension of the data. Finally, they provide some empirical results on synthetic data demonstrating their ability to recover the intrinsic dimension with a conservative vector field, and an inability to recover intrinsic dimension with a non-conservative vector field.

**Strengths:**

This paper studies an important and relevant problem and derives a simple, clear, condition to precisely characterize when a vector field leads to the same marginal distributions as the true score. Using this equation, they cleanly show that conservativity is neither sufficient nor necessary, theoretically answering a question which has been studied empirically in the literature.

**Weaknesses:**

The derivation that leads to their main finding, the gauge invariance in equation (11), is not especially novel, since it follows immediately from the definition of the reverse time process. Also, their main finding - if I understand correctly, that gauge invariance as in (11) is necessary and sufficient for the time marginals of the learned process to be identical to those of the true process - is not formally stated as a mathematical result. I think the paper would be helped by a clear and precise mathematical statement of this main finding.

Also, Theorem 1 on the intrinsic dimension estimation problem states that conservativity implies their intrinsic dimension estimation procedure is consistent (under their assumptions). But their gauge invariance finding (equation (11)) states the law of the reverse time process is unaffected under this gauge invariance, so it would be more natural to expect that the intrinsic dimension estimation procedure is consistent under the gauge invariance (so that conservativity is not necessary). And then in the discussion about Figure 3, the authors claim that their empirics demonstrate that conservativity leads to accuracy of the intrinsic dimension estimation procedure, while models which aren't constrained to be conservative have inaccurate intrinsic dimension estimation. But this is an odd finding give that their main observation is this gauge invariance, so that in particular, we would expect that enforcing conservativity is not important. There is no formal contradiction here, since the trained models which aren't enforced to be conservative may not satisfy the gauge invariance, but it does seems to somewhat contradict the essence of the paper. Finally, the empirics in Figure 3 (they are difficult to read so it's possible I am mis-interpreting them) seem to be inconsistent with their discussion of them and their Theorem 1 (see below).

**Questions:**

- I suggest you formalize the gauge invariance principle
- As mentioned above, the relationship of the empirics, Theorem 1, and the gauge invariance principle are confusing. It would be good to study and discuss more how these results complement (or don't) one another.
- For Figure 3: I don't see why the true singular values must be 1. They should depend on the local geometric properties of the manifold, the true data density, and the map.
- For Figure 3: the colors are very difficult to make out. Please use fewer lines (e.g. throw out 2 of the 3 non-material singular values) and more visual differentiation (e.g. different line types).
- For Figure 3: as it is constructed (and from what I can tell due to the difficult of making out the details, see above), it seems there are more than 2 singular values (green, black, yellow) converging to 1, which seems to contradict Theorem 1 and the discussion in the first paragraph after "Intrinsic dimensionality estimation using diffusion models" where it is said "2 of them converge to 1, whereas the remaining 3 diverge"
- There are many typos.

---

> ### Author Response · Authors · 2023-11-19
> **Response to reviewer pZCv**
>
> We thank the reviewer for the many intriguing questions and suggestions. We will answer them in order of appearance:
>
> + Indeed, we agree that a formal statement would be appropriate. Please refer to our official comment: https://openreview.net/forum?id=92KV9xAMhF&noteId=0bECf5o20B.
>
> + Please refer to our official comment: https://openreview.net/forum?id=92KV9xAMhF&noteId=0JS67DC6Z8.
>
> + We agree that the relationship between the empirics and the gauge freedom requires some further thought, especially if one solely understands them separately as a way to estimate the intrinsic dimensionality (which is a difficult task on its own) and as a principle to understand under which conditions a diffusion model is exact, respectively. In principle, a remainder term fulfilling the gauge freedom conditions must also lead to the exact intrinsic dimensionality - we agree with the reviewer on that.
> However, from a practical point of view, fulfilling the gauge freedom conditions is challenging, and the nature of the diffusion model $s_{\theta}$ will affect the trajectories approaching the data manifold. This is exactly what we observed in the experiments: even though a conservative $s_{\theta}$  does not match $s$ exactly, the trajectories approaching the data manifold are sufficiently well-behaved to allow the estimation of the intrinsic dimensionality. This observation and the new corollary (a) (which states that the gauge freedom term $r$ is $0$ if and only if the vector field is conservative) suggest enforcing conservativity. This establishes a new link between theorem 1 and the implications from section 3.
>
> + The reviewer correctly states that the true singular values are not generally 1 but depend on the local geometric properties, the true data density, and the map. However, in this case, the manifold is a 2-D plain, the true data density a Gaussian, and the map a simple padding. Therefore, the singular values are indeed 1.
>
> + We agree with the unfortunate choice of colors. We apologize for this inconvenience and will accommodate it in the camera-ready version.
>
> + Indeed, this misconception is due to our unfortunate choice of colors and line types. Only two converge to 1. We apologize for this inconvenience and will update the figure following the reviewer's suggestions.
>
> + We will carefully proofread to reduce the number of typos. We thank the reviewer for pointing this out.

---

> > ### Comment · Reviewer_pZCv · 2023-11-20
> > **Please add brief comment about the true singular values**
> >
> > Thank you for your response, I have updated my score accordingly.
> >
> > Could you please also add a brief comment about the true singular values in Figure 3? I believe you are using here the precise form the Ornstein-Uhlenbeck process initialized at a (singular) Gaussian. This is not obvious (to me, at least).

---

> > > ### Author Response · Authors · 2023-11-20
> > > **Brief comment on the true singular values**
> > >
> > > Thank you for updating your score, and for your curious question.
> > >
> > > The easiest way to understand why the singular values must be 1 is through Figure 2. There, you can see how the singular values of $Y_t$ evolve as a function of time. Note that these singular values are the (positive) eigenvalues of $Y_t Y_t^T$. As we show in equation (16), those eigenvalues precisely describe the local variability of a small neighborhood of samples (depicted as a dotted circle around $\mathbf{x}_t$). In Figure 2, the data manifold is a circle and the on-manifold distribution is a Von Mises distribution. However, in the experiment underlying the left-hand side of Figure 3, the manifold is a 2-D plain and the distribution is a standard 2-dimensional Gaussian. Therefore, the local variability (described by the eigenvalues of $Y_t Y_t^T$) must be 1.
> > >
> > > Does this explanation make more sense?

---

> > > > ### Comment · Reviewer_pZCv · 2023-11-21
> > > > **Please add a rigorous proof to the appendix or remove the claim**
> > > >
> > > > That doesn't seem to be a rigorous proof. Please add a completely rigorous proof or simply remove the claim from your paper. It isn't necessary for the rank = 2 claim anyways.
> > > >
> > > > Note that, in general, there are many ways to flow one distribution into another. You need to be using the specific structure of this flow and the true data distribution if you want to make a claim of this nature.

---

> > > > > ### Author Response · Authors · 2023-11-21
> > > > > **Removing the claim**
> > > > >
> > > > > We thank the reviewer for insisting on a formal argument. Our argument given before was more an intuition and by no means a  rigorous proof. Indeed, formal proof requires calculations involving the specific structure of the flow. Our intuition might just coincide with the simple nature of the example. As pointed out by you, it is not necessary for our ID estimator and we will therefore remove the claim from the paper.

---

### Official Review · Reviewer_ovjk · 2023-10-30

**Soundness:** 3 good
**Presentation:** 2 fair
**Contribution:** 2 fair
**Rating:** 5
**Confidence:** 4

**Summary:**

This paper investigates an additional degree of freedom that appears in the approximate learnt score function in a diffusion model when one relaxes the constraint that the score function must be a conservative vector field.

They do this by analysing the modified score function $s(x, t) = \nabla \log p_t(x) + r(x, t)$.

They show that under the constraints of 1) $\nabla \cdot r(t, \cdot) = 0$, $r$, is conservative and 2) $\nabla \log p_t(x) \cdot r(x, t)=0$, $r$, is orthogonal spatially to the score, that when initialised with the same initial distribution, the evolution of the distribution of the SDE evolved through time with and without the additional $r$ term remains the same.

They then apply reasoning from this conclusion ideas around density estimation, data generation, and local information estimation.

**Strengths:**

The topic discussed is an interesting one, and would have impact on how people train score based models in practise for the specific task of dimension estimation.

The presentation of diffusion models in section 2 is clean.

The main result of section 3 is interesting.

**Weaknesses:**

- While the analysis in section 3 is interesting, there is no sense of how one could exploit these conditions in practise.
- The first half of section 4 to me does not make sense from a practical perspective (see questions)
- The experimentation to back up section 5 is limited.

**Questions:**

- Do the authors have an intuitive interpretation of the conditions in 12?
- How can one practically realise the constraints of equation 12? Imposing the orthogonality constraint in combination with the divergence free part seems as difficult as just estimating the exact score.
- If one takes no drift (i.e. the ode does not move the distribution), then it becomes clear that the constraints force the dynamics to be such that 1) particles are constrained move along the same level-set of the distribution (i.e. through time a particle remains at $p_t(x)=c$ for all $t$), the orthogonal constraint, as long as 2) the dynamics do not alter the density of those level sets, the divergence free constraint. Is this perspective interesting when combined with a non-zero drift?
- I do not understand the point of the first part of section 4. I understand the argument that given a path $x_t$, if one adds a divergence free term to the score preserved density estimation. However, why would one assume they have a path? Without the exact score, or one satisfying the additional orthogonality constraint, how would one be able to generate a correct path from the model?
- Section 4.1, it would be highly instructive of the authors to plot examples of the trajectories with and without the additional $r$ term I think. From my own plots, one ends up with a series of trajectories that at each instance are "spinning" around the mean, following the shape of the covariance ellipses. One can multiply the $r$ term by a constant to speed up said spin. This related to my previous point also. Even if one initialises the ODE with the reference measure, the particles can still move as they can follow the covariance ellipses at constant speed on each ellipse.
- The experiments of section 5 do not seem to be related to theorem 1. Theorem 1 says that you can correctly estimate the data dimensionality if the extra $r$ field satisfies the gauge freedom constraints. However the experiments only use a fully unconstrained score, and a score constrained to be conservative. There is not experiment showing that one can estimate the dimension correctly if the vector field is non-conservative, but does satisfy the gauge constraints.

---

> ### Author Response · Authors · 2023-11-19
> **Response to reviewer ovjk**
>
> We thank the reviewer for the many intriguing questions and remarks. We will answer them in order of appearance:
>
> + Because of the very insightful comments of reviewer h3ie, we will drop the individual conditions (see old equations (12a) and (12b)) while only focusing on the gauge freedom condition (see eq. (10)). Indeed, the orthogonality perspective provides a very intuitive interpretation of the gauge freedom condition.
>
> + Please refer to the general answer: https://openreview.net/forum?id=92KV9xAMhF&noteId=0JS67DC6Z8.
>
> + Indeed, it is an interesting observation and provides intuition for the old equations (12a) and (12b).
>
> + Indeed the first part of Section 4 assumes a correct path is given, which cannot be tested without knowing the exact score. This observation should be understood as a motivation for remark 1.2 where we explain the difference between a gauge freedom for the instantaneous change of variable formula and a gauge freedom for the Fokker-Planck equation. We think it is not apparent why only the latter yields correct sampling and density estimation, and therefore, we think the reader will benefit from this observation.
>
> + This is a great idea and would provide additional intuition. We thank the reviewer for this suggestion.
>
> + We don't agree with the reviewer on that point. Theorem 1 does not say that the data dimensionality can be estimated if the extra field satisfies the gauge freedom constraints. Indeed, the gauge freedom is simply an assumption. We show in Theorem 1 that if in addition, the score is conservative, then the intrinsic dimensionality can be estimated. Thus, conservativity is a sufficient condition (given all the other assumptions from the Theorem). Therefore, it makes sense to compare a conservative vs. a non-conservative vector field using Theorem 1 to estimate the intrinsic dimensionality.

---

### Official Review · Reviewer_h3ie · 2023-10-31

**Soundness:** 4 excellent
**Presentation:** 4 excellent
**Contribution:** 3 good
**Rating:** 8
**Confidence:** 5

**Summary:**

This paper investigates whether learning a conservative vector field is a necessary and/or sufficient condition to perform various generative modeling tasks, namely, sampling, density estimation, and dimensionality estimation. For the first task (and maybe the second one, see below), the authors show that it is slightly different condition which is necessary and sufficient. For the third task, the authors show that being conservative is sufficient.

**Strengths:**

The paper tackles an important issue on which there is both interest and confusion in the community, as there have been several conflicting studies on the topic of conservativeness of learned score vector fields. Section 3 is particularly well-written and makes an important point that had been overlooked in the literature. The numerical experiments in Section 5 are also very convincing.

**Weaknesses:**

I have several comments:

- Though the $r$ part of the learned score does not influence the marginals along the ODE trajectory in _continuous time_, it might do so when the ODE is _discretized_. In that case, it seems reasonable that "straighter" trajectories would be easier to discretize, and that these would correspond to $r = 0$, thus giving another practical motivation for regularizing the network to be conservative.

- I am slightly confused by section 4. It seems to me that a subtle point when evaluating densities is that the score $s$ appears both explicitly in the integral in (3), but also implicitly through the trajectory (3). Therefore, it seems hard to evaluate exactly the effects on the modeled density. Besides, it seems that the paper make conflicting statements about the result: is the condition that $r$ satisfies (10), or that it satisfies both (12a) and (12b) (which are stronger than (10))? Different parts of the paper state different results (e.g., abstract and Figure 1 vs section 4). It seems to me that $r$ satisfying (10) is sufficient, since the samples have the correct distribution, and the change of variable formula still holds for $s$, so (3) must compute the correct density.

- Theorem 1 should state that the assumptions imply that $r$ vanishes when $t \to 0$ (as shown in the proof), which makes it easier to understand the theorem. Also, the assumption that the divergence is finite may be questionable (why would $r$ remain stable when $t \to 0$ when the score explodes?), and should be discussed. In the proof, I don't see why the finite divergence of $r$ should imply that $r_\theta^T\nabla \log p_t \to 0$. Fortunately, these problems can be solved, the assumption removed, and the proof simplified. In fact, as it stands, the theorem is misleading, as assuming that $s_\theta$ is conservative implies that $s_\theta = s$ and $r =0$, and it is thus a statement about the true score rather than an approximation of it.

- Let me suggest two additional references. [1] studies the approximation power of unconstrained conservative neural networks (not parameterized as the gradient of a scalar function). Essentially, exact conservativeness implies that the network only depends on a one-dimensional projection its input [1, Theorem 6]. It explains why $r$ will never be exactly zero in practice, unless one explicitly parameterizes the score as the gradient of a scalar function. This should be mentioned in the introduction together with the literature review on the topic. [2] studies the intrinsic dimensionality of image manifolds using the singular values of the network Jacobian and is thus very much related to Section 5. An observation that might be of interest to the authors is that for natural images the dimensionality _increases_ when the noise level vanishes, as opposed to what one expects for a low-dimensional manifold.

[1] Saremi, Saeed. “On Approximating $\nabla F$ with Neural Networks.” arXiv, November 6, 2019. http://arxiv.org/abs/1910.12744.

[2] Mohan, Sreyas, Zahra Kadkhodaie, Eero P. Simoncelli, and Carlos Fernandez-Granda. “Robust and Interpretable Blind Image Denoising via Bias-Free Convolutional Neural Networks.” arXiv, February 8, 2020. http://arxiv.org/abs/1906.05478.

**Questions:**

I also have been thinking about score conservativeness for a while. I thus make several suggestions which I hope can help the authors improve the paper (which is already interesting and quite clear!).

### On Helmholtz decompositions

I did not find equations (12a) and (12b) to be helpful, but rather confusing. Rather than the Helmholtz decomposition, the right decomposition is that any vector field $s \in L^2(p_t)$ can be decomposed uniquely as $s = \nabla \phi + r$ where $\phi$ is a scalar field and $r$ satisfies the gauge freedom equation (10) (this decomposition then depends on $p_t$). Furthermore, this decomposition is orthogonal, in the sense that $\mathbb E[||s(x_t)||^2] = \mathbb E[||\nabla \phi(x_t)||^2 + ||r(x_t)||^2]$. This is readily checked with the integration by parts identity $\mathbb E[\nabla \phi(x_t) \cdot r(x_t)] = -\mathbb E[\phi(x_t) \left( \nabla \cdot r(x_t) + \nabla \log p_t(x_t) \cdot r(x_t) \right)]$ for any vector field $r$ and scalar field $\phi$, which shows that conservative fields $\nabla\phi$ are orthogonal in $L^2(p_t)$ to vector fields $r$ which satisfy (10). More abstractly, the adjoint of the gradient operator $\nabla$ is the "gauge freedom" operator $-(\nabla + \nabla \log p_t) \cdot$, showing that the image of the former is the orthogonal complement of the kernel of the latter.

I believe that stating this decomposition helps making the paper clearer. In particular, it has several direct implications for the paper:
- Because of the orthogonality property, the score-matching loss can be decomposed $\mathbb E[|| \nabla \log p_t(x_t) - s_\theta(x_t) ||^2] = \mathbb E[|| \nabla \log p_t(x_t) - \nabla \phi(x_t) ||^2 + ||r(x_t)||^2]$. This shows that (denoising) score matching training naturally regularizes the $r$ part of the score to have a small norm (but it cannot be zero, see above). We also see that $s_\theta$ is conservative if and only if $r=0$.
- It simplifies the proof of Theorem 1 and removes an unnecessary assumption. Indeed, the assumption that $s_\theta$ is conservative is equivalent to $r = 0$ (at all times!). There is thus no need to assume that the divergence of $r$ is finite.

### Typos

- "scaler" bottom of page 2
- "satisfie" middle of page 6
- "boundery", top of page 8
- "symmyrizing", bottom of page 8
- "conservity", middle of page 9
- "freemdom", bottom of page 9

---

> ### Author Response · Authors · 2023-11-19
> **Response to reviewer h3ie**
>
> We are very thankful for the reviewer's insightful comments. In particular, we fully agree that the orthogonality perspective makes the message much more straightforward. We will, therefore, update the derivation of the gauge freedom conditions and present them as the condition required to obtain an orthogonal decomposition of the vector field; see our general comment: https://openreview.net/forum?id=92KV9xAMhF&noteId=0bECf5o20B. As pointed out by the reviewer, the consequence is that gauge freedom conditions and the conservativity condition imply that $r = 0$. We will update the text and Figure 3 accordingly.

---

> > ### Comment · Reviewer_h3ie · 2023-11-20
> >
> > Thank you for your response. I am glad my comments helped improve the paper.

---

### Official Review · Reviewer_KRsj · 2023-11-01

**Soundness:** 2 fair
**Presentation:** 3 good
**Contribution:** 2 fair
**Rating:** 8
**Confidence:** 4

**Summary:**

This paper explores the conservativeness of the score vector field in the context of diffusion models. The author show that density estimation and sampling can be exact even if the learned score vector field is non-conservative, as long as the gauge freedom condition holds. This condition enforces that the remainder vector field (i.e. the gap between the true and learned vector fields) must be both divergence-free and orthogonal to the true score everywhere. Two dimensional empirical examples are presented to support the derivations. Moreover, the authors show that for exact estimation of of intrinsic dimensionality of the underlying manifold, a conservative vector field is sufficient.

**Strengths:**

The paper is written fairly clearly with a nice flow. The topic of conservativeness of the learned vector field in the context of diffusion models or denoising is still an open-ended question, with important empirical and theoretical implications. Proposing the gauge freedom condition as a valid relaxation of conservativeness requirement is an interesting and fairly significant theoretical contribution.

**Weaknesses:**

In my opinion, the main weakness of the paper is that it cannot yield to real world practical results. More specifically, in order to test or enforce the gauge freedom condition (eq 11 or conditions 12a and 12b), one needs to have access to the true score, which is not generally possible. In fact, diffusion models are mostly used for cases where the true score is not known. So it is not clear how knowing this condition would solve the question of conservativeness. The empirical results given in the paper are all toy examples for this very reason.

The result of section 4 states that for exact density estimation only condition 12a needs to be satisfied. In other words, if the discrepancy between the true score and the learned score can be written as a rotation then the density can be estimated precisely from eq 3. However, this result also assume that the model generates exact samples, which is not clear how can be tested in real cases such as image densities.

**Questions:**

In section 4.1 a two dimensional example is discussed to show that conservativeness of score is neither necessary nor sufficient. The true score in this example has a divergence of zero which make it quite trivial. Is this an inevitable consequence of conditions 12a and 12b in 2 dimensions? In other words, doesn't satisfaction of gauge freedom conditions imply that the true score must have zero divergence in 2D? Since the authors use this example to make a more general conclusion in higher dimensions, it might be more convincing if 1) it is shown whether an example exists where the true score divergence is not zero yet the conditions are satisfied 2) if such example exists, that could replace the current example so that the generalization to higher dimensions would be more realistic.

---

> ### Author Response · Authors · 2023-11-19
> **Response to reviewer KRsj**
>
> We thank the reviewer for the constructive review. It seems that the reviewer's concerns are twofold. First, the practical implementations of the gauge freedom findings as it is necessary to have access to the true score. Second, the example in section 4.1 is trivial in 2D as the gauge freedom condition implies zero divergence in 2D.
>
> Regarding the first concern, we refer to our general answer: https://openreview.net/forum?id=92KV9xAMhF&noteId=0JS67DC6Z8.
>
> Regarding the second concern. We disagree with the reviewer's statement, "The true score in this example has a divergence of zero, which makes it quite trivial." Actually, the divergence term of the true score $s(\mathbf{x},t)$ is equal to $-\sigma_{1}^{-2} - \sigma_{2}^{-2}$, which is not $0$ as both $\sigma_{1}$ and $\sigma_{2}$ are assumed to be positive. Therefore, we think this example serves as a non-trivial counter-example for conservativity not being necessary for exact sampling and density estimation. In case this does not answer your question, please don't hesitate to clarify.

---

> > ### Comment · Reviewer_KRsj · 2023-11-21
> >
> > Thank you for your response. I agree that there is great value in stating gauge freedom condition even without immediate practical implications and it could open avenues for empirical exploration of the problem in the future. I also realized after reading your comments that assuming zero divergence for the 2d Gaussian example was incorrect. That example could indeed be very useful for building intuition in the special case where stronger conditions 12a and 12b hold.
> >
> > I agree with reviewer h3ie that removing equations 12a and 12b will simplify the text a lot and makes it less confusing. Although it can still be useful as a special simple case of condition 11 for building intuition (for example in the 2d Gaussian example).
> >
> > I really appreciate the many constructive comments by my co-reviewers and think the paper has great potential to convey your important message about conservativeness in a clear way, assuming the suggested comments will be incorporated in the final draft. Thus I raise my score to 8.

---

### Author Response · Authors · 2023-11-19
**General answer**

We thank all the reviewers for carefully reading our manuscript. We are very grateful for the many comments, suggestions, and interesting questions. In particular, the comments of reviewer h3ie will help to substantially improve our manuscript by providing great intuition on the gauge freedom condition (more details in the official comment "formalization of the gauge freedom condition" below).

Additionally to the formalization of section 3, we provide a global response to the recurrent question "on the practical feasibility". For the remaining questions and remarks, we respond to the reviewers directly.

---

> ### Author Response · Authors · 2023-11-19
> **Formalization of the gauge freedom condition**
>
> Reviewer pZCv suggested formalizing the first part of section 3 as a theorem, while reviewer h3ie suggests replacing the Helmholtz decomposition with a more general decomposition (see below). We think that the excellent remark of reviewer h3ie naturally leads to a formalization of section 3 and an overall simplification of different parts of the paper. In particular, as mentioned by reviewer h3ie, theorem 1 can be simplified, and the manifold gauge freedom conditions (12a) and (12b) can be removed. In particular, section 3, together with the reviewer's h3ie suggestions, lead to the following theoretical conclusions:
>
> *Decomposition Theorem:* Every vector field in $s\in L^2(p_t)$ can be decomposed uniquely as $s=\nabla \phi +r$ where $\phi$ is a scalar field and $r$ satisfies the gauge freedom condition (10).
>
> *Proof sketch:* Reviewer h3ie shows that indeed the "gauge freedom" operator $-(\nabla + \nabla \log p_t)$ is the adjoint of the gradient operator $\nabla$. This shows that the set of all vector fields fulfilling the gauge freedom condition (10) is the closed, orthogonal complement of the set of all conservative vector fields in $L^2(p_t)$. The uniqueness of the projections onto these two subspaces follows from the Hilbert projection theorem.
>
> The Decomposition Theorem has some direct implications:
>
> *Corollaries:* Let $s=\nabla \phi +r$.
>
> (a) If $s$ is conservative, then it must hold that $r=0$ (this was shown by reviewer h3ie).
>
> (b) $s$ provides exact samples and density estimation if and only if $\nabla \phi = \nabla \log p_t$
> (this follows from the uniqueness of the decomposition and the calculations from section 3).
>
>
> We will add this theorem and the corollaries to the end of section 3 for the camera-ready version while removing the conditions (12a) and (12b).

---

> ### Author Response · Authors · 2023-11-19
> **On the practical feasibility of the gauge freedom condition**
>
> Indeed, it isn't easy to practically realize the gauge freedom condition expressed in equation (10). However, the same is true for classical diffusion models at first glance. It is indeed surprising that the score-matching objective, which involves only the known conditional transition probability $\nabla p_{0t} (\mathbf{x_{t}} | \mathbf{x_0})$, leads to learning the true unknown score $\nabla \log p_{t}(\mathbf{x})$.
> A potential Gauge freedom regularization aims to learn a scalar product $r_{\theta}^{T}\nabla \log p_{t}(\mathbf{x})$, which is different from simply learning the true score $\nabla \log p_{t}(\mathbf{x})$, and thus it is not apparent if a score-matching ansatz (replacing $\nabla \log p_{t}(\mathbf{x})$ with $\nabla \log p_{0t}(\mathbf{x_{t}} | \mathbf{x_0})$) will yield to a remainder term fulfilling the gauge freedom condition. However, we also think it is *not* obvious that such or a similar ansatz will *not* work. Therefore, our work opens new exciting research questions for the diffusion community.

---

> > ### Author Response · Authors · 2023-11-21
> > **A few more thoughts on the relevance of the gauge freedom condition.**
> >
> > Even though we currently do not know how to practically test whether the remainder term $r_\theta = s_\theta - \nabla \log p_t$ of a specific vector field $s_\theta$  satisfies the gauge freedom condition, it does not mean that this gauge condition is irrelevant. And this is for several reasons.
> > 1. First, the fact that we currently do not know how to test a given condition practically does not mean that it will never be possible to test it directly or indirectly. Actually, we believe it could be an interesting research question (see our previous comment https://openreview.net/forum?id=92KV9xAMhF&noteId=0JS67DC6Z8)
> > 2. Secondly, as highlighted by reviewer pZCV, the gauge freedom condition can be understood from an orthogonality perspective (see also  https://openreview.net/forum?id=92KV9xAMhF&noteId=jwbeNMqn9r). This provides a conceptual framework that helps to reason about conservative vector fields and deviations from conservativity. This result (that we will formulate as a theorem) also has practical consequences - that we will formulate as corollaries (see https://openreview.net/forum?id=92KV9xAMhF&noteId=0bECf5o20B)
> > 3. Finally, it should be noted that gauge freedom in Physics had a huge impact  - despite the fact that we can not “test” a specific gauge (such as the Coulomb or the Lorentz gauge in electromagnetism). A specific gauge's relevance is found in the extent to which the (electromagnetic) equations can be simplified in a given situation. Similarly, for diffusion models, one could expect that the gauge freedom condition could be exploited to simplify some relevant equations/statements. For example, as noted by reviewer pZCV, the gauge freedom condition enables to re-express the score matching loss as a sum of a conservative loss and a loss for the remainder term (see https://openreview.net/forum?id=92KV9xAMhF&noteId=jwbeNMqn9r).

---

### Meta-Review · Area_Chair_CDkt · 2023-12-12

**Metareview:**

The paper studies the issue of conservativity in diffusion models. In diffusion models, the denoising step is motivated as an approximation to the score function, i.e., the gradient of the log likelihood. The vector field defined by the score function is conservative, in the sense that it is the gradient of some smooth function. Vector fields implemented by general neural networks (including neural net approximations to the score function) are not conservative; there has been much discussion in the literature about the effect of this on learned diffusion models.

The main observation of the paper is that conservativity is not necessary for exact sampling or density estimation. Rather, a weaker condition, which the paper calls the gauge freedom condition, needs to be fulfilled. Roughly, this implies that there is a rotational degree of freedom — in addition to propagating probability mass along the direction of the score function, we are free to rotate mass around this direction.

The paper also makes observations on intrinsic dimension estimation — showing that with a conservative vector field, the intrinsic dimension is correctly estimated from the rank of the Jacobian of the vector field, whereas with nonconservative vector fields, intrinsic dimension can be overestimated.

**Justification For Why Not Higher Score:**

The paper provides theoretical clarity on an issue that has received much discussion in the experimental literature on diffusion models. At the same time, its direct computational implications are unclear, since the gauge freedom condition depends on the underlying true score function.

**Justification For Why Not Lower Score:**

The paper clarifies that the conservativity condition is not necessary for exact sampling and density estimation in diffusion models; it also argues that conservativity is needed for intrinsic dimension estimation. It introduces an alternative condition -- gauge freedom, which is necessary and sufficient for exact sampling. This resolves an issue which has been much discussed in the experimental literature, with (previously) inconclusive results.

---

### Decision · Program_Chairs · 2024-01-16

Accept (poster)